# Indirect questioning methods for sensitive survey questions: Modelling criminal behaviours among a prison population

**Beatriz Cobo[1], Eva Castillo[2,3], Francisca López-Torrecillas**  **[3,4], María del Mar Rueda**  **[1]**\*

**1** Department of Statistics and Operational Research, University of Granada, Granada, Spain, **2** Granada – Albolote Penitentiary Center, Granada, Spain, **3** Mind, Brain, and Behavior Research Center (CIMCYC), University of Granada, Granada, Spain, **4** Department of Personality, Evaluation and Psychological Treatment, University of Granada, Granada, Spain

\* mrueda@ugr.es

## Abstract

Information such as the prevalence and frequency of criminal behaviour is difficult to estimate using standard survey techniques because of the tendency of respondents to withhold or misrepresent information. Social desirability bias is a significant threat to the validity of self-reported data, especially when supplied by persons such as sexual offenders or those convicted of theft or substance abuse. The randomized response approach is an alternative to the standard interview method and offers great potential for researchers in the field of criminal justice. By means of a survey of 792 prison inmates, incorporating both indirect and direct response techniques, we investigate if the prison population also has problems recognizing their participation in criminal acts such as theft, illicit drug use, violence against property, reckless driving and arson. Our research findings suggest that self-reported criminal behaviour among a prison population is affected by social desirability bias and that the behaviour considered is significantly associated with the severity of obsessive-compulsive symptoms. The results also demonstrate the inadequacy of traditional, yet widely used, direct questioning methods, and the great potential for indirect questioning techniques to advance policy formation and evaluation in the field of criminal behaviour.

## Introduction

The prison population is growing and researchers have highlighted the need for specific, reliable treatment measures to reduce the incidence of criminal behaviour such as illicit drug use, sexual aggression, theft and dangerous driving. According to Cerezo [1] most inmates are sentenced for drug-related crimes (37.9% of cases), property-related crimes (35.3%) or homicide/ assault (12%). The European Drug Report (EMCDD, [2]) noted that 21% of prisoners recognised having used cannabis while in prison and 0.4%, drugs by intravenous administration. Significant numbers of prisoners had substance abuse or addiction problems, involving heroin (14%), cocaine (27%), alcohol (31%) or cannabis (40%), together with associated problems, especially HIV and hepatitis C. In many cases, too, drug addiction provokes depressive symptoms, autolysis, irritability and physical and/or psychological suffering. These symptoms are often correlated with a past history of family violence or sexual abuse and co-occur with crimes

**Data Availability Statement:** There are restrictions for the publication of the data set. The data contains information about crimes and penalties. Data can be requested from the Vice-Rector's

Office for Research and Scientific Policy, email:
investigacion@ugr.es.

**Funding:** This work is partially supported by
Ministerio de Economía y Competitividad of Spain
(grants MTM2015-63609-R and PID2029-
106861RB-I00).

**Competing interests:** The authors have declared
that no competing interests exist.

such as sexual aggression and theft, as well as with traffic offences. Furthermore, these statistics are computed on the basis of a single custodial sentence per crime, i.e. only the crime considered to be the most serious is recorded for analysis, although an individual may have been sentenced in more than one respect [3].

The existence of repetitive, harmful behaviour has been attributed to traits traditionally described as "impulsive" or "compulsive", such as substance dependence, gambling addiction or hoarding. These situations are common and often co-occur, both among the general population and among prison inmates [4,5]. However, to our knowledge the possible association between many forms of criminal behaviour and the characteristics of impulsivity and compulsivity has not been addressed in previous research. The ground-breaking nature of this study is reflected in this is the first time that the construct of compulsivity will be defined in prison populations and delimited differences among impulsivity and compulsivity. This study is the first time that impulsivity and/or compulsivity in men who commit violent crimes.

A prominent problem in criminology is that of understanding what determines deviant and/or illegal behaviour. The question of why some people commit crimes while others remain law-abiding is associated with the nature and impact of motivation and with how institutions can influence behaviour. To explain such behaviour variations, precise estimates are needed, but due to the sensitive nature of this subject, criminal behaviour is difficult to study empirically, and valid information is scarce. There is widespread concern that self-reported offending measures are an imprecise measure of delinquency [6,7] Questions such as the prevalence and frequency of criminal behaviour are difficult to estimate using standard survey techniques because respondents tend to withhold information.

A relevant factor is that of social desirability bias (the desire to make a favourable impression on others), which poses a significant threat to the validity of self-reports. This is particularly the case with persons such as sexual offenders, those who commit robbery with violence and those with problems of substance abuse. In this type of inquiry setting, the randomized response (RR) approach constitutes an alternative to traditional interview methods, and offers great potential to researchers in the criminal justice field. By using RR methods, studies of the prevalence of illegal phenomena can be conducted more ethically and can yield more valid estimates [8,9].

Our study aim is to measure the prevalence of certain forms of criminal behaviour–theft, illicit drug use, violence against property, reckless driving (i.e speeding) and arson–among a Spanish prison population. To do so, we use a RR method, applied according to individual circumstances, including age, level of education and psychological characteristics. This investigation addresses the following specific question:

*Research Question*: Does the population of prison inmates in Granada (Spain) present social desirability bias about certain forms of criminal behaviour as theft, illicit drug use, violence against property, reckless driving (i.e speeding) and arson?

Various questioning methods have been devised to ensure respondents' anonymity and to reduce the incidence of evasive answers and the over/underreporting of socially undesirable acts. These methods are generally known as indirect questioning techniques and they obey the principle that no direct question is posed to survey participants, whose privacy thus remains protected.

There exist various types of indirect questioning strategy for eliciting sensitive information, including randomized response, item count and non-randomized response. RR was the first to be formulated and has been the object of most theoretical and empirical study. With this technique, a randomization device is used to determine whether the respondent should answer the sensitive question or another, neutral one, irrespective of true status concerning the sensitive behaviour. The principle underlying this method is that if respondents believe their answer

does not disclose their reality to the interviewer, they will be more likely to give accurate information about behaviour of a sensitive nature.

Since the pioneering study [10] many RR mechanisms have been proposed and analyzed. Existing methods have been improved, and new ones proposed [e.g. 11–14]. RR techniques have been used in many empirical studies addressing different forms of illegal behaviour, such as the use of illicit drugs [e.g. 15–21] Others have considered the prevalence of abortion [e.g. 22–24] or of sexual assault [e.g. 25, 26], the illegal use of natural resources [e.g. 27, 28], the non-compliance with Dutch regulatory laws [e.g. 29], the reception of stolen goods [e.g. 30], female genital ablation [e.g. 31], corruption in Olympic sports [e.g. 32] or gender violence [e.g. 33, 34]. There are few works in which RR methods are used to address behaviour involving theft, violence against property, reckless driving or arson. Some of these are [35] where employee theft is investigated or the study of [36] about offenses, including vandalism, drug use, rape, arson, and robbery in a population of students in sociology courses.

The novelty of this work with respect to other RRT works lies in the investigated population: the prison population. The social desirability bias is expected to be observed in people who follow the rules and comply with the laws, but in prison populations, where criminal behavior is the order of the day, and some are very frequent (such as drug use) has not been investigated and it could be questioned whether, despite being very common behaviors, this population has problems in openly acknowledging its practice.

## Materials and methods

### Ethics statement

The researchers and the entities that have collaborated (Albolote Penitentiary Center, Granada) have strictly complied with Law 14/2007 on Biomedical Research and Organic Law 15/1999 on Protection of Personal Data. This research has been carried out in accordance with the European Union, national, and regional legislation covering the use of human data for scientific purposes.

All participants were given the names of the principal investigators, emails, Departments, and Research Center involved, as well as a clear description of the research objectives. All participant enrolled in the present study signed informed consent before their inclusion.

Before starting the research, the study had been approved by the Ethics Committee of the University of Granada and by the Spanish Ministry of the Interior.

### Participants

This study was conducted by means of a survey of inmates at the prison in Granada (Spain). The only criteria for inclusion were that respondents should be willing to participate and be aged 18–55 years. Persons with physical or mental illness (such as schizophrenia or depression) or currently receiving psychopharmacological treatment were excluded. Following application of these criteria, 792 men were included in the study group. The respondents had a mean age of 37.08 years (SD = 9.27).

The participants were interviewed individually. Those who met the inclusion criteria were then invited to participate. Those who accepted completed the Symptom Inventory (SCL-90-R; [37]) and then took part in an individual interview, as described below. At the beginning of every session, they were informed about the study aims and reminded that they had the right to abandon the study at any moment. Every respondent provided signed informed consent to participate. All the participants could read and write. During the study measurements, the researchers answered the questions that the participants had. At the end of each session, the participants were debriefed and thanked for their participation.

## Measures

**Demographics, criminal record and institutional behaviour interview.** The interview, designed specifically for this project, is intended to obtain sociodemographic data, information regarding the crime for which the prison sentence is being served and details of the prison sentence received, in accordance with applicable legislation [38].

The sociodemographic distribution of the study population is shown in Table 1. The data obtained were analyzed for the whole study population and also according to the variables education, marital status, nationality, crime committed and in-prison conduct. The table also shows the quantitative variables considered: age, the prison term imposed (in months) and the scores recorded for the obsession and compulsion (mean and standard deviation).

**The Checklist of Yale-Brown Obsessive-Compulsive scale (Y-BOC; [39]).** The obsessive-compulsive scale is designed to provide a detailed description of obsessions and compulsions, divided into forty symptom dimensions. These include obsessions about harm due to aggression/injury/violence/natural disaster; sexual/moral/religious obsessions and related compulsions; obsessions about symmetry or 'just-right' perceptions; compulsions to count or order/arrange; obsessions regarding contamination and/or cleaning; obsessions and compulsions related to hoarding; and miscellaneous obsessions and compulsions related to somatic concerns and superstitions.

**Table 1. Sociodemographic distribution of the sample.**

| Variables | Scores |
|---|---:|
| **Education qualifications** | |
| None | 125 |
| Primary | 377 |
| Secondary | 207 |
| Higher | 83 |
| **Marital status** | |
| Single | 376 |
| Married | 160 |
| Divorced | 113 |
| Cohabiting | 141 |
| **Nationality** | |
| Spanish | 747 |
| Other | 45 |
| **Crime** | |
| Theft | 335 |
| Violence | 119 |
| Sex crime | 52 |
| Homicide, murder | 49 |
| Other | 235 |
| **In-prison conduct** | |
| Model | 449 |
| Acceptable | 301 |
| Wrong | 42 |
| **Age** | 37.01 (9.317) |
| **Length of sentence (months)** | 69.76 (72.626) |
| **Obsessive-compulsive score** | 7.34 (7.121) |

The scale is based on a 64-item measure of obsession and compulsion severity, recorded according to the situation 'last week', each of which is scored from 0–4. A score of zero is assigned when no such problems are reported. Scores of 1–4 reflect mild, moderate, severe and very severe obsession/compulsion states, respectively. The questionnaire items pertain either to obsession or compulsion, and the scores for each are first examined to calculate the Obsession and Compulsion Severity Scales, separately. All items are then summed to calculate the Total Severity Score, which is categorized as low (8–15 points), moderate (16–23 points), severe (24–31 points) or very severe (32–40 points). In addition, ten items reflecting the severity of obsession/compulsion are assessed on a five-point scale ranging from 0 (no symptoms) to 4 (extremely severe symptoms) with respect to time spent, interference, distress, resistance and control. Thus, the total score awarded in this respect ranges to 0 to 40 points.

This scale presents an acceptable degree of validity and reliability [40]. One study of 40 respondents recorded an inter-evaluator reliability of 98% and an internal consistency coefficient (alpha coefficient) of 89% [41]. Another recorded an inter-evaluator consensus of 0.86 and an internal consistency coefficient of 0.88 (0.95 in the Spanish-language version) [42].

## Study variables

In the present study, the participants were asked to indicate whether they had ever engaged in behaviour involving theft, illicit drug use, violence against property, reckless driving (speeding) or arson. The following questions (here, translated from the Spanish) were asked:

1. Have you ever stolen?

2. Have you ever consumed illegal drugs?

3. Have you ever committed violent acts against property?

4. Have you ever driven dangerously fast?

5. Have you ever committed arson?

## Statistical analysis

Randomized response, a technique first proposed [10] is used to protect respondents' privacy when sensitive questions must be answered. In RR, two questions are posed and the respondent is asked to answer one or the other depending on the outcome of a randomizing device.

In the present study, a more advanced procedure, often used in current practice, was applied: the Forced Response Design (FRD). In this approach [43] the person $i$ is offered a box with cards: some are marked "Yes" with a proportion $p_1$, some are marked "No" with a proportion $p_2$ and the rest are marked "Genuine", in the remaining proportion $p_3 = 1 - p_1 - p_2$, where $0 < p_1, p_2 < 1$. The person is requested to randomly draw one card and to respond 1 if the card is marked "Yes", 0 if it is marked "No" and to give the true answer if the card is marked "Genuine".

The Horvitz-Thompson estimator of the true proportion of the sensitive variable, its variance and the confidence intervals were calculated with the *ForcedResponse* function of the R package RRTCS [44].

In the present study, multivariate regression analysis, with logistic regression, was also performed using GLMMRR package in R software [45]. The logistic regression model for FRD can be viewed as a particular case of the generalized linear model for RR proposed [46] with parameters $c = (1 - p_1)p_2$, $d = p_1$ and the logit link function.

## Procedure

In our survey, both direct questioning (DQ) and an indirect questioning approach (FRD) were employed. The respondents were randomly assigned to one of these two methods. The survey was conducted in the prison, where inmates are not allowed to use any type of electronic device. Accordingly, the randomization mechanism consisted of a deck of cards. This was a Spanish-format deck, consisting of 40 cards, divided into four suits, each numbered from one to seven, plus three figures. Each inmate was instructed in the procedure as follows:

1. Select a card.

2. If the card chosen is number 1 or 2, your answer to the question must be "Yes", regardless of the true answer. If it is number 3 or 4, your answer must be "No". If it is 5, 6 or 7 or a figure, please answer the question honestly.

3. Do not tell the interviewer which card you have chosen (to maintain your anonymity about the answers given).

4. Return the card to the deck and repeat the process for the other questions.

## Results

### Direct versus indirect response methods

The point estimates for the (sensitive) study variables and the corresponding 95% confidence intervals for each technique (DQ and RR) are summarized in Table 2.

For the first variable, *Theft*, the indirect estimate obtained is higher than the direct estimate, and the difference is statistically significant. This finding reflects social desirability bias, i.e. respondents' tendency to answer according to their understanding of what is socially acceptable. Similar results were obtained for the other study variables (*drugs*, *violence against property*, *reckless driving* and *arson*). In every case, higher values are obtained by the indirect technique than by the direct method, and the differences are statistically significant.

We also include a measure for the effect size. Cohen's d values show "medium" effects size for all considered variables except for *reckless driving* whose effect size is "small". This may be due to the extremely high prevalence of this behavior which is a problem for the estimation with this technique.

**Table 2. Estimated prevalence of the behaviours considered.**

| Variable | Method | Estimation | Variance | Lower bound | Upper bound | P-value DQ vs RR | Cohen's d |
|---|---|---|---|---|---|---|---|
| **Theft** | *Direct* | 0.5762 | 0.0005 | 0.5287 | 0.6238 | <0.001 | 0.26 |
| | *Indirect* | 0.8329 | 0.0014 | 0.7592 | 0.9066 | | |
| **Drugs** | *Direct* | 0.7021 | 0.0005 | 0.6581 | 0.7461 | <0.001 | 0.23 |
| | *Indirect* | 0.9983 | 0.0010 | 0.9338 | 1.0000 | | |
| **Violence** | *Direct* | 0.3002 | 0.0005 | 0.2561 | 0.3443 | <0.001 | 0.53 |
| | *Indirect* | 0.5867 | 0.0016 | 0.5068 | 0.6666 | | |
| **Speeding** | *Direct* | 0.7215 | 0.0004 | 0.6784 | 0.7646 | <0.001 | 0.1 |
| | *Indirect* | 0.9418 | 0.0012 | 0.8731 | 1.0000 | | |
| **Arson** | *Direct* | 0.0750 | 0.0001 | 0.0497 | 0.1004 | <0.001 | 0.74 |
| | *Indirect* | 0.2679 | 0.0015 | 0.1907 | 0.3451 | | |

## Sub-populations

In addition to obtaining results for the sensitive variables for the entire population, we also obtained them for specific categories: education, marital status and in-prison conduct, as shown in Figs 1–3, for the case of the sensitive variable theft.

These figures show that for the variable "*Theft*", the social desirability bias is statistically significant when the inmates have only primary education, are single and whose in-prison conduct is either model or acceptable.

The graphs for the rest of the sensitive variables can be seen in Figs 4–6.

For the variable "*Drugs*", the differences are significant for the inmates with primary or secondary education, for all types of marital status and for those whose in-prison conduct is either model or acceptable.

For the variable "*Violence against property*", the differences are significant for all levels of education except university studies, for inmates who are single or cohabiting and for all categories of in-prison conduct.

For the variable "*Speeding*", the differences are significant for inmates with primary or secondary education, for all types of marital status except those who are married and for those whose in-prison conduct is either model or acceptable.

For the variable "*Arson*", the differences are significant for inmates with only primary education, for all types of marital status except "divorced" and for those whose in-prison conduct is either model or acceptable.

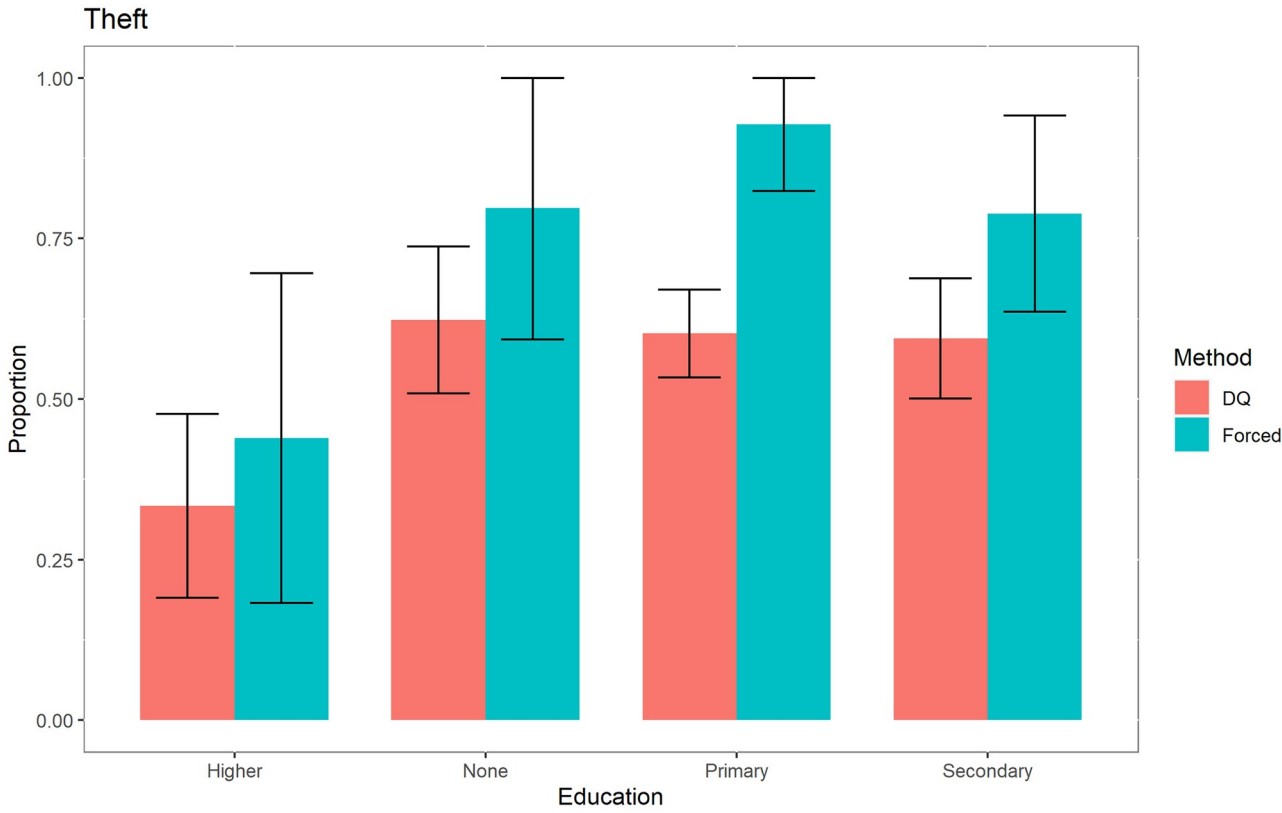

**Fig 1. Prevalence of theft according to education.**

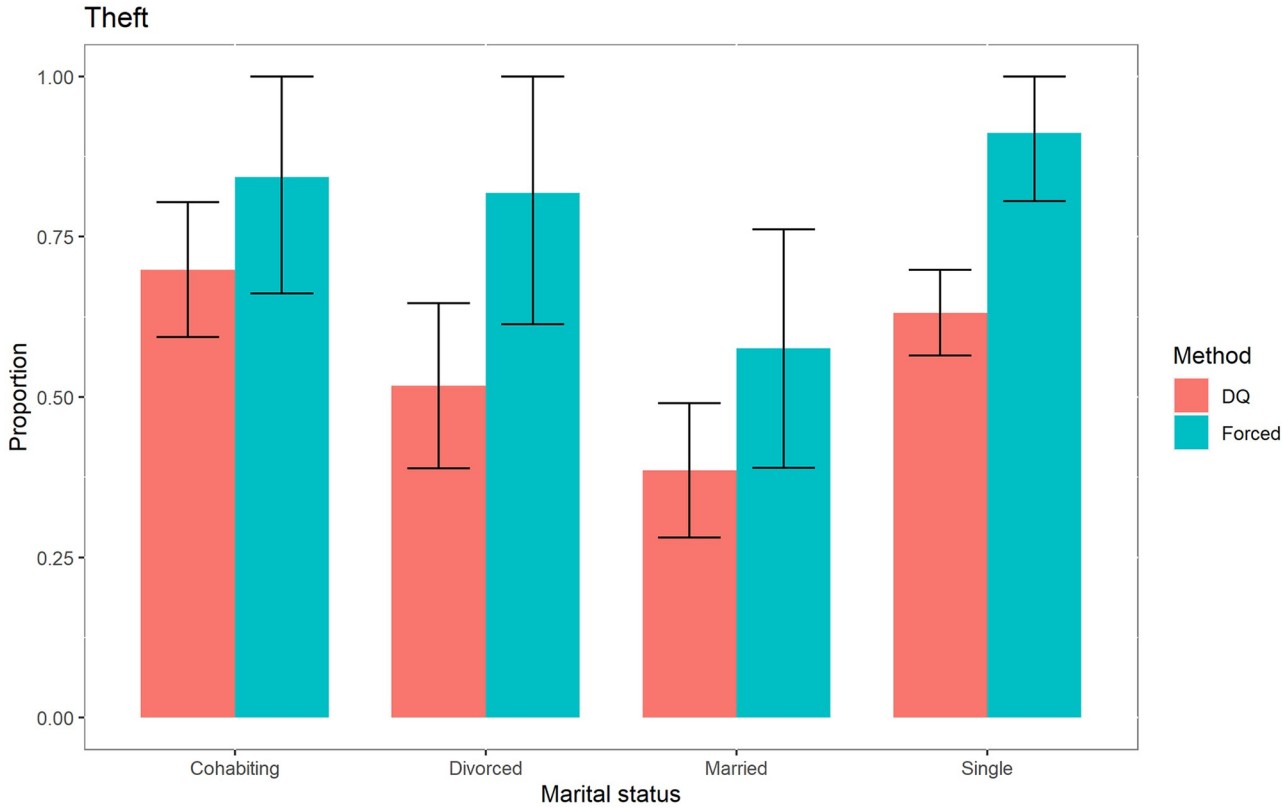

**Fig 2. Prevalence of theft according to marital status.**

## Regression

A multivariate analysis of the response data was carried out to investigate the effects of the questioning technique and the background variables. The following explanatory variables were included in the models: questioning method, education, marital status, crime, in-prison conduct, age, length of sentence (in months) and obsessive-compulsive score. To obtain the best regression model, the variables were selected by a step-wise procedure based on the Akaike Information Criterion (AIC). We consider the Pearson statistic as the goodness-of-fit statistic. The Wald test score shows us which variables are significant in the model. The coefficients and the corresponding standard errors obtained are shown in Tables 3–5. The reference classes for the qualitative variables are Direct questioning (*Method*), No formal qualifications (*Education*), Single (*Marital status*), Theft (*Crime*) and Model (*In-prison conduct*).

We present a logistic model only for the variables theft, drugs and arson since results were stable for these variables.

Our results show that inmates questioned with the forced method are more likely to admit to theft and that the odds ratio increases by a factor of 10.1443 for those who are questioned via the forced method, compared to those who are questioned directly. As regards marital status, the results show that married prisoners are less likely to admit theft than those who are single. Prisoners whose in-prison conduct is wrong are more likely to admit to theft than those whose conduct is model. The variables "*Sentence imposed*" and "*Obsessive-compulsive score*" are positively associated with "*Theft*". In other words, if the sentence is increased by one month, the probability of the respondent admitting theft rises by 0.65%. Similarly, if the

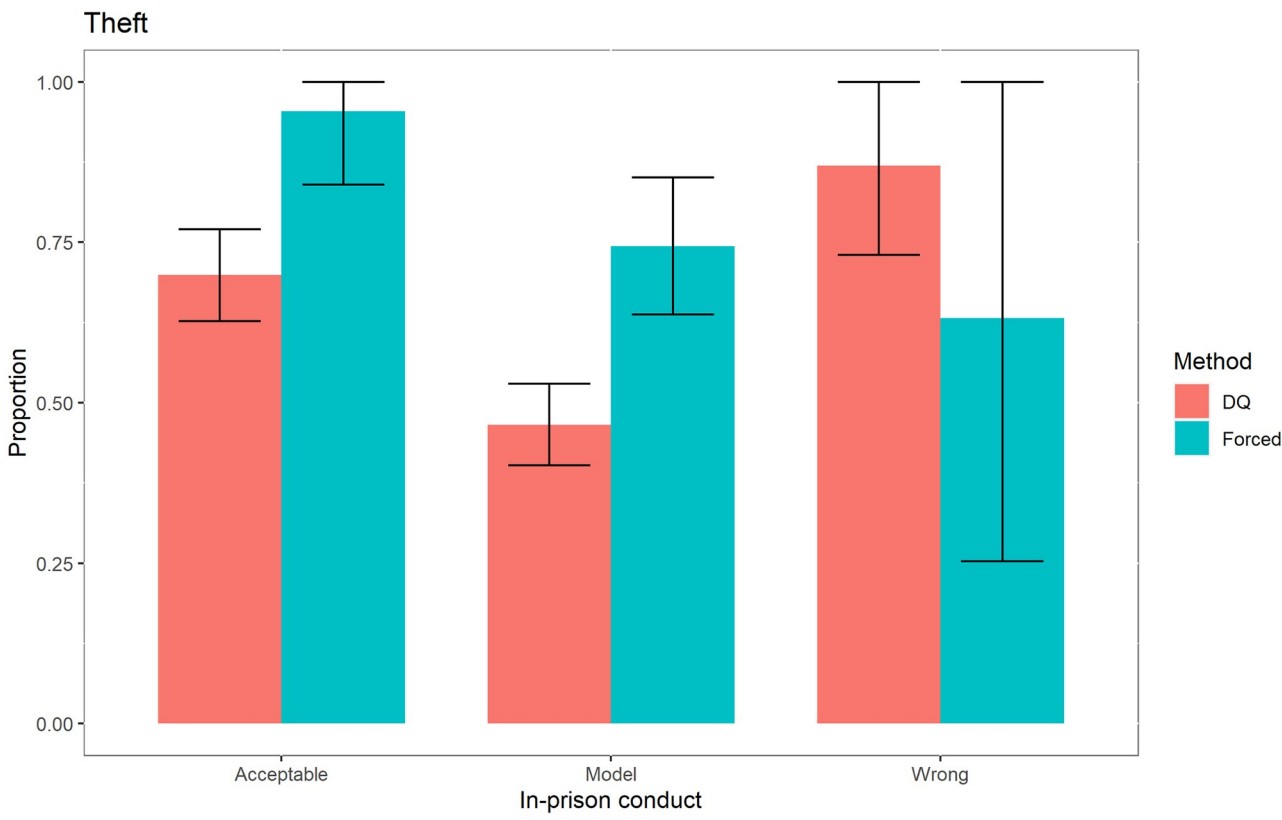

**Fig 3. Prevalence of theft according to in-prison conduct.**

obsessive-compulsive score increases by one unit, the probability of the respondent admitting theft rises by 6.03%.

Application of the same analysis to the variable "*Drugs*" reveals the following. The odds ratio increases, with a factor of approximately 207 for the use of FRD versus DQ. Divorced men are less likely to admit to drug consumption than those who are single. This is also true for the highly educated versus those with no formal education. Inmates whose conduct is acceptable or wrong are more likely to admit to drug consumption than those whose conduct is model. There is a positive association between the presence of obsessive-compulsive score and the recognition of drug consumption.

Table 5 shows that with respect to "*Arson*", the odds ratio is approximately 14 times higher for inmates questioned via the forced method than for those questioned directly. A positive relationship was observed between obsessive-compulsive score and arson, i.e. inmates who present a higher degree of obsession-compulsion are more likely to admit their involvement in arson.

## Discussion

In this study we compared the use of direct and indirect questioning methods to investigate the following forms of behaviour among a population of prison inmates: theft, illicit drug use, violence against property, reckless driving (speeding) and arson.

The values obtained by the indirect approach were found to be higher than those obtained directly, and these differences are statistically significant. This difference reflects the existence

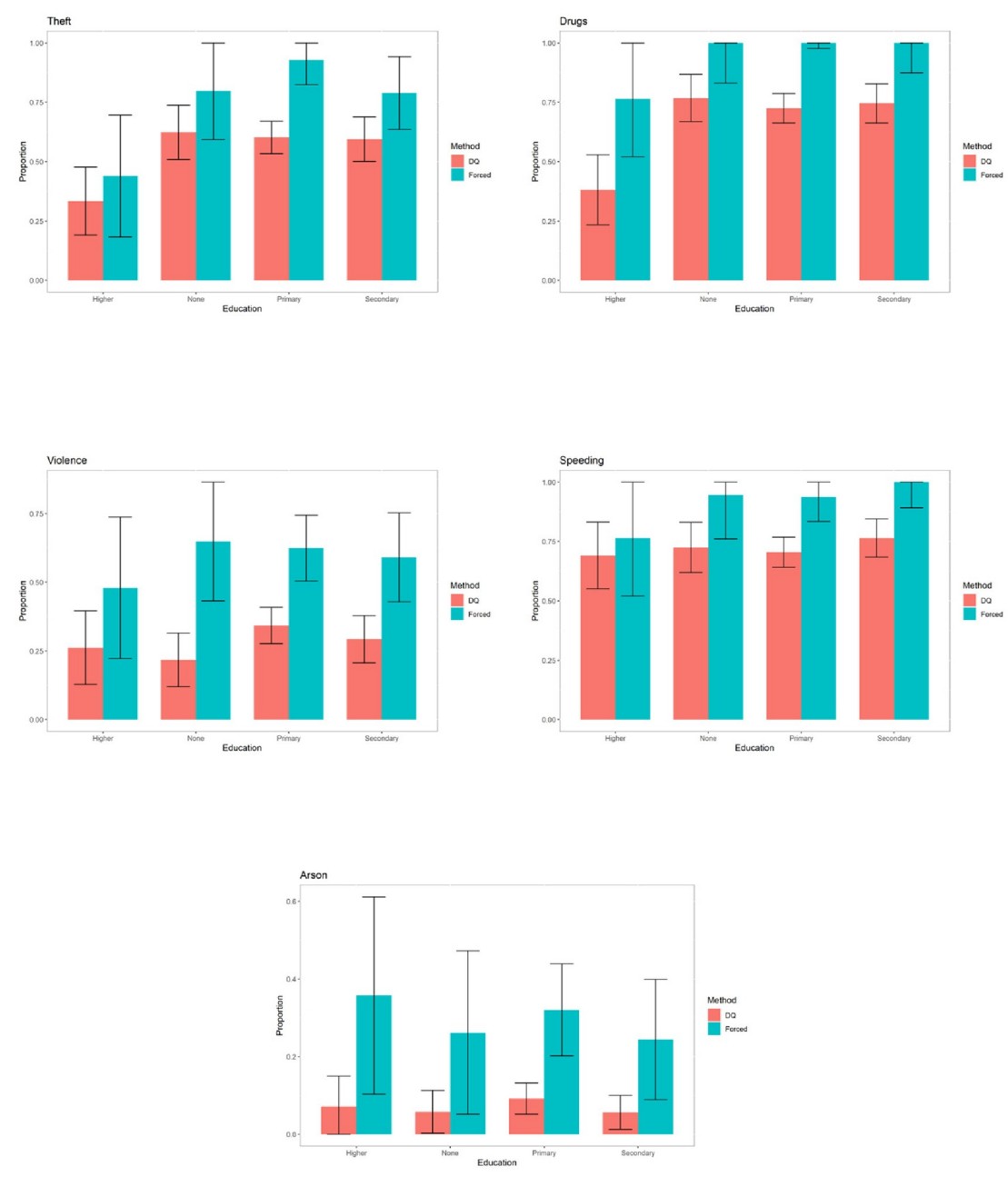

**Fig 4. Prevalence of sensitive variables for specific categories of education.**

of social desirability bias, i.e., the tendency of respondents to answer in accordance with their belief as to what is considered socially acceptable. These findings are in line with other authors [9] who argued that social desirability and the fear of sanction may deter respondents from giving truthful answers to sensitive questions. Self-reports on norm-breaking behaviour such as theft, illicit drug use, violence against property, speeding and arson may thus lead to significant under-estimation, resulting, among other problems, in the distortion of population statistics. Our results show that the RR technique reduces this kind of bias.

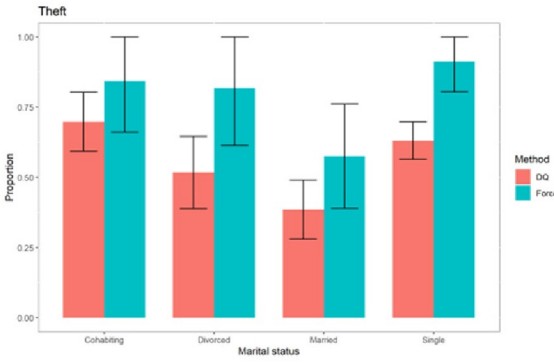

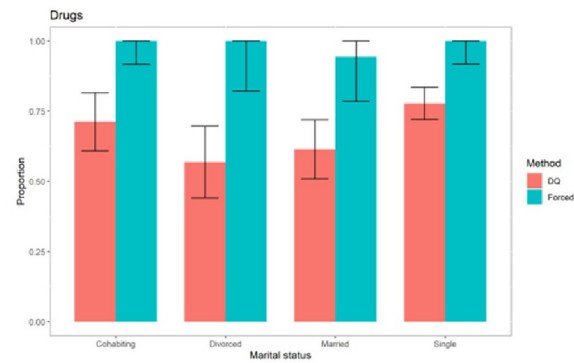

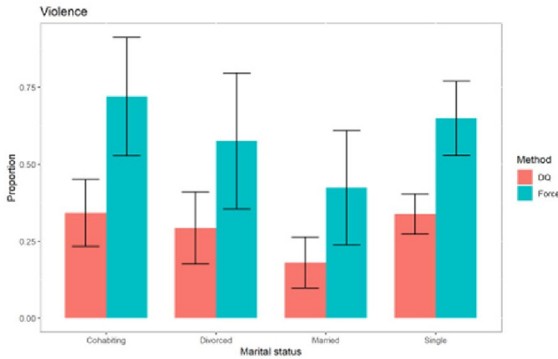

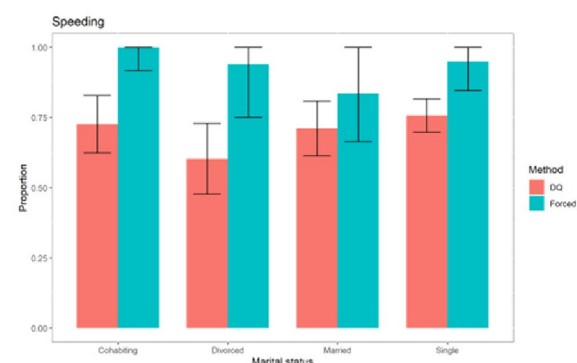

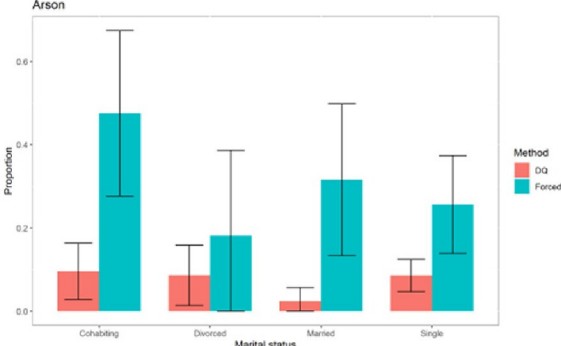

**Fig 5. Prevalence of sensitive variables for specific categories of marital status.**

Our study results also show that the RR method enables the prevalence of behaviour patterns to be classified according to variables such as the crime committed and the inmate's education, marital status and in-prison conduct. For example, the indirect questioning method produces significantly higher responses than the direct method for the study variable, *theft*,

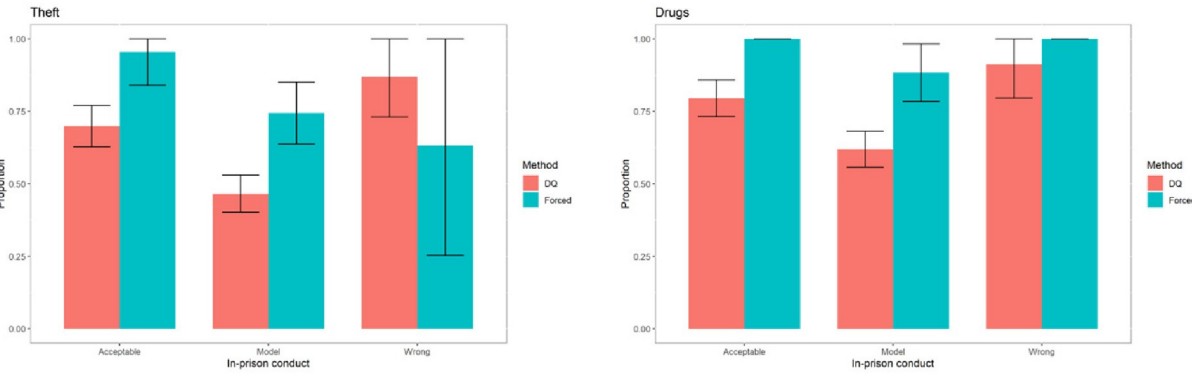

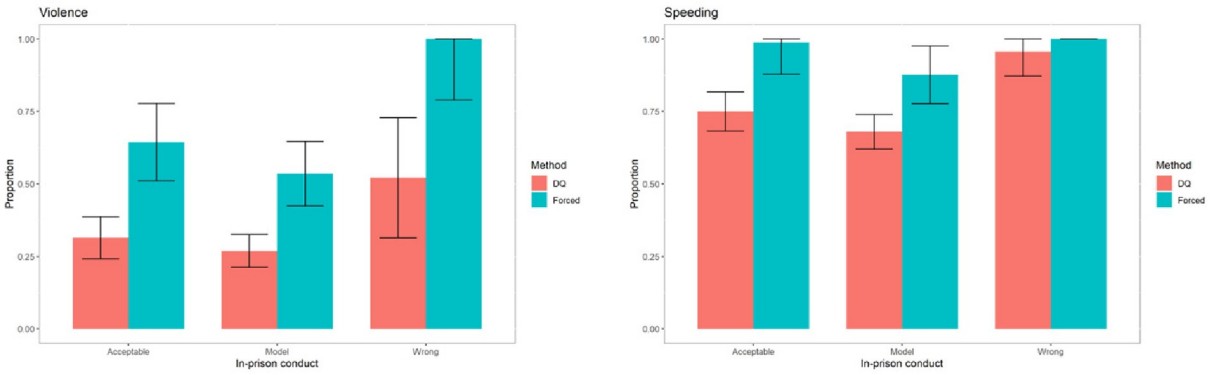

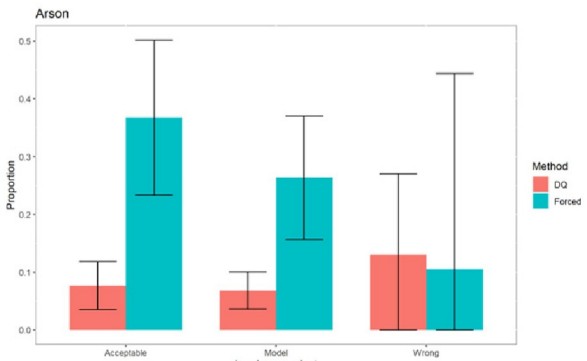

**Fig 6. Prevalence of sensitive variables for specific categories of in prison conduct.**

**Table 3. Logistic regression coefficients estimated by multivariate regression analysis for "*Theft*".**

| Variable | Category | Estimate | Std. error | Exp(estimate) | P.value |
|---|---|---|---|---|---|
| **Theft** | | | | | |
| **(Intercept)** | | 0.5930 | 0.3010 | 1.8095 | 0.0488* |
| **Method** | *Forced* | 2.3169 | 0.4140 | 10.1443 | <0.001*** |
| **Marital status** | *Married* | -0.7398 | 0.3083 | 0.4772 | 0.0164* |
| | *Divorced* | -0.0104 | 0.3422 | 0.9896 | 0.9757 |
| | *Cohabiting* | 0.3850 | 0.3350 | 1.4697 | 0.2504 |
| **Crime** | *Violence* | -1.7879 | 0.3518 | 0.1673 | <0.001*** |
| | *Sex crime* | -3.0138 | 0.4929 | 0.0491 | <0.001*** |
| | *Homicide, murder* | -2.2686 | 0.5164 | 0.1035 | <0.001*** |
| | *Other* | -1.8188 | 0.2981 | 0.1622 | <0.001*** |
| **In-prison conduct** | *Acceptable* | 0.4243 | 0.2697 | 1.5286 | 0.1156 |
| | *Wrong* | 1.9446 | 0.6880 | 6.9911 | 0.0047** |
| **Sentence (m)** | | 0.0065 | 0.0022 | 1.0065 | 0.0032** |
| **Obsessive-Compulsive score** | | 0.0585 | 0.0173 | 1.0603 | 0.0017** |
| AIC | 901.23 | | | | |
| | P.value | | | | |
| Pearson | 0.56092 | | | | |

Signif.codes:

'***' 0.001,

'**' 0.01,

'*' 0.05,

'.' 0.1.

when the inmate has only primary education, is single and whose in-prison conduct is either model or acceptable.

These results represent new understanding in this field. To our knowledge, no previous studies have been undertaken to investigate the relationship between crime, education, marital status and in-prison conduct with the existence and degree of social desirability bias. Our findings show that the RR technique is more effective than direct questioning for eliciting truthful opinions about socially undesirable behaviour, specifically theft, illicit drug use, violence against property, reckless driving and arson. Furthermore, the degree of truthfulness in the responses given varies according to the respondents' education, marital status and in-prison conduct.

The estimated logistic regression coefficients derived from our multivariate regression analysis produced the following results. Married prisoners are less likely than those who are single to admit to "*Theft*". However, this finding must be qualified when in-prison conduct is taken into account; inmates whose conduct is wrong are more likely to admit to theft than those whose conduct is model. Moreover, our analysis reveals a positive association between length of sentence, obsessive-compulsive score and theft. These findings corroborate [4,5] and suggest that pathological impulsivity and compulsivity characterize a broad range of criminal behaviours.

Divorced inmates are less likely than those who are single to admit to having consumed illicit drugs. Similarly, those who are highly educated are less likely to admit it than those with no educational qualifications. However, inmates whose in-prison conduct is wrong or only acceptable are more likely to admit to the consumption of drugs than those whose conduct is model. Obsessive-compulsive score is positively associated with this variable; thus, inmates

**Table 4. Logistic regression coefficients estimated by multivariate regression analysis for "*Drugs consumption*".**

| Variable | Categories | Estimate | Std. error | Exp(estimate) | P.value |
|---|---|---|---|---|---|
| **Drugs** | | | | | |
| **(Intercept)** | | 1.2960 | 0.4049 | 3.6545 | 0.0014** |
| **Method** | *Forced* | 5.3366 | 2.5239 | 207.8008 | 0.0345* |
| **Marital status** | *Married* | -0.4495 | 0.3232 | 0.6380 | 0.1643 |
| | *Divorced* | -0.7704 | 0.3546 | 0.4628 | 0.0298* |
| | *Cohabiting* | -0.5358 | 0.3409 | 0.5852 | 0.1161 |
| **Education** | *Primary* | -0.4209 | 0.3543 | 0.6564 | 0.2348 |
| | *Secondary* | -0.2257 | 0.3933 | 0.7980 | 0.5661 |
| | *Higher* | -1.5319 | 0.4717 | 0.2161 | 0.0012** |
| **Crime** | *Violence* | -0.7216 | 0.3649 | 0.4860 | 0.0480* |
| | *Sex crime* | -1.5822 | 0.4779 | 0.2055 | 0.0009*** |
| | *Homicide, murder* | -0.9157 | 0.5196 | 0.4002 | 0.0780 |
| | *Other* | -0.5373 | 0.3078 | 0.5843 | 0.0809 |
| **In-prison conduct** | *Acceptable* | 0.7571 | 0.2717 | 2.1322 | 0.0053** |
| | *Wrong* | 1.6734 | 0.7908 | 5.3302 | 0.0343* |
| **Obsessive-compulsive score** | | 0.0724 | 0.0194 | 1.0750 | 0.0002*** |
| AIC | 855.07 | | | | |
| | P.value | | | | |
| Pearson | 0.820 | | | | |

Signif.codes:

'***' 0.001,

'**' 0.01,

'*' 0.05,

'.' 0.1.

**Table 5. Logistic regression coefficients estimated by multivariate regression analysis for "*Arson*".**

| Variable | Categories | Estimate | Std. error | Exp(estimate) | P.value |
|---|---|---|---|---|---|
| **Arson** | | | | | |
| **(Intercept)** | | -3.0526 | 0.3134 | 0.0472 | <2e-16*** |
| **Method** | *Forced* | 2.6975 | 0.2769 | 14.8419 | <2e-16*** |
| **Crime** | *Violence* | -0.8972 | 0.4341 | 0.4077 | 0.0388* |
| | *Sex crime* | 0.9746 | 0.4917 | 2.6501 | 0.0475* |
| | *Homicide, murder* | -1.1933 | 0.6850 | 0.3032 | 0.0815. |
| | *Other* | 0.0095 | 0.3071 | 1.0096 | 0.9752 |
| **Obsessive-compulsive score** | | 0.0673 | 0.0184 | 1.0696 | 0.0002*** |
| AIC | 709.31 | | | | |
| | P.value | | | | |
| Pearson | 0.4788 | | | | |

Signif.codes:

'***' 0.001,

'**' 0.01,

'*' 0.05,

'.' 0.1.

with a higher obsessive-compulsive score are more likely to admit to drug consumption. To our knowledge, no previous research study has addressed this area. Nevertheless, drug consumption is known to be prevalent among the prison population [2,5] and compulsivity and impulsivity undoubtedly play a role in addictive behaviours. Accordingly, it is important to identify a framework within which to conceptualize and separate impulsive and compulsive problem behaviours, in order to explore common or distinct antecedents.

The study variable "*Arson*" is positively associated with obsessive-compulsive score. This finding, too, is novel. However, our results are consistent with other authors [47] who emphasized the need to examine the evidence for its effectiveness, and discuss new directions to enhance it as therapy for obsessive-compulsive behaviour.

In conclusion, in this study, we estimate the proportion of prison inmates who have engaged in certain forms of criminal behaviour (theft, illicit drug use, violence against property, reckless driving and arson). Our findings suggest that inquiries into these types of behaviour may be subject to social desirability bias. Moreover, these behaviours are significantly related to the severity of obsessive-compulsive scores.

Our results demonstrate the inadequacy of traditional, yet widely used, direct questioning methods, and highlight the great potential offered by indirect questioning techniques to obtain a more accurate evaluation of sensitive topics and thus advance policy formation and evaluation in the field of criminal behaviour.

## Acknowledgments

The authors would like to thank the participants, as well as the Granada–Albolote Penitentiary Center, especially thanks to Enrique Gomez Sánchez for his help with recruiting participants.

## Author Contributions

**Conceptualization:** Francisca López-Torrecillas.

**Data curation:** Eva Castillo.

**Formal analysis:** Francisca López-Torrecillas.

**Methodology:** Beatriz Cobo, María del Mar Rueda.

**Software:** Beatriz Cobo.

**Supervision:** María del Mar Rueda.

**Validation:** Eva Castillo.

**Writing – original draft:** María del Mar Rueda.

**Writing – review & editing:** Francisca López-Torrecillas.

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
