## [Decision Letter · Decision Letter 0]

24 Sep 2020

PONE-D-20-23586

Indirect questioning methods for sensitive survey questions: modelling criminal behaviours among a prison population

PLOS ONE

Dear Dr. Rueda,

Thank you for submitting your manuscript to PLOS ONE. After careful consideration, we feel that it has merit but does not fully meet PLOS ONE’s publication criteria as it currently stands. Therefore, we invite you to submit a revised version of the manuscript that addresses the points raised during the review process.

The reviewers have serious doubts about some aspects of the article that the authors should consider. Reviewers' comments are complementary. Reviewer 1 is very critical of the literature review, the purpose of the research question and shows some statistical concerns. Reviewer 2 focuses primarily on statistical aspects of research question 3. Authors should carefully consider these issues.

The reviewers have serious doubts about some aspects of the article that the authors should consider. Reviewers' comments are complementary. Reviewer 1 is very critical of the literature review, the purpose of the research question and shows some statistical concerns. Reviewer 2 focuses primarily on statistical aspects of research question 3. Authors should carefully consider these issues.

We look forward to receiving your revised manuscript.

Kind regards,

José J. López-Goñi

Academic Editor

PLOS ONE

Journal Requirements:

2. Please provide additional information regarding the considerations  made for the prisoners included in this study. For instance, please discuss whether participants were able to opt out of the study and whether individuals who did not participate receive the same treatment offered to participants.

3.We note that you have indicated that data from this study are available upon request. PLOS only allows data to be available upon request if there are legal or ethical restrictions on sharing data publicly. For more information on unacceptable data access restrictions, please see http://journals.plos.org/plosone/s/data-availability#loc-unacceptable-data-access-restrictions.

4.We note that [Figure(s) 5 - 45] in your submission contain [map/satellite] images which may be copyrighted. All PLOS content is published under the Creative Commons Attribution License (CC BY 4.0), which means that the manuscript, images, and Supporting Information files will be freely available online, and any third party is permitted to access, download, copy, distribute, and use these materials in any way, even commercially, with proper attribution. For these reasons, we cannot publish previously copyrighted maps or satellite images created using proprietary data, such as Google software (Google Maps, Street View, and Earth). For more information, see our copyright guidelines: http://journals.plos.org/plosone/s/licenses-and-copyright.

1.    You may seek permission from the original copyright holder of Figure(s) [5 - 45] to publish the content specifically under the CC BY 4.0 license. 

5. Please include a caption for figure 5 - 45.

Additional Editor Comments (if provided):

The reviewers have serious doubts about some aspects of the article that the authors should consider. Reviewers' comments are complementary. Reviewer 1 is very critical of the literature review, the purpose of the research question and shows some statistical concerns. Reviewer 2 focuses primarily on statistical aspects of research question 3. Authors should carefully consider these issues.

Reviewers' comments:

Reviewer's Responses to Questions

**Comments to the Author**

1. Is the manuscript technically sound, and do the data support the conclusions?

Reviewer #1: Partly

Reviewer #2: Yes

2. Has the statistical analysis been performed appropriately and rigorously? 

Reviewer #1: Yes

Reviewer #2: No

3. Have the authors made all data underlying the findings in their manuscript fully available?

Reviewer #1: No

Reviewer #2: No

4. Is the manuscript presented in an intelligible fashion and written in standard English?

Reviewer #1: Yes

Reviewer #2: Yes

5. Review Comments to the Author

Reviewer #1: The article “Indirect questioning methods for sensitive survey questions: modelling criminal behaviours among a prison population” is in my opinion not strong enough to deserve publication in Plos One.

1. It shows that the randomized response technique does what it always does, revealing a higher frequency of admitting having done various crimes than direct questioning does. The article does not give us a reason why we should have doubted this outcome for this special case (prison inmates, a number of crimes that according to the authors have not yet been researched). So what is new here? It is completely in line with eg the meta-analysis of Lensvelt et al.

2. Why should we be interested in criminal prevalence for theft, violence against property, reckless driving, arson in a population incarcerated for theft, violence, sex crimes, homicide murder, “other”?

a. The article does not analyse the special subgroups that are incarcerated for theft or violence, which may be interesting as we know the true behaviour (if we trust the Spanisgh judges).

b. It is not true that no studies have been done on RR for arson (Durham & Lichtenstein) or theft (eg Wimbush & Dalton).

3. The authors tell us that no studies have been done on the relation crime – impulsivity. That is completely wrong, indeed it is a very active field in criminology, following Wilson & Hernstein; Gottfredson).

4. While logistic regression results are given, we do not get much interpretation out of them.

a. Should you not compare and comment on the size of the various coefficients compared over tables (eg what to make out of the result that “married” is significant in table 3 and insignificant in all other tables?).

b. What about effect sizes? Is “obsessive/compulsive” an important factor, with exp(estimates) of 1.04 to 1.07? Or is this negligible.

c. You should comment on an estimate of 13731595.8244. It points to such a skew distribution of the dependent that the logistic regression is probably numerically instable. (Figure 44 shows us that indeed a 100% prevalence in some categories, which is a problem for the algorithm).

5. The paper is unduly long.

6. All in all, I think this research may deserve a small research note, but certainly not a full blown article in Plos One.

Durham, A. M., & Lichtenstein, M. J. (1983). Response Bias in Self-Report Surveys-Evaluating Randomized Responses (From Measurement Issue in Criminal Justice, P 37-57, 1983, Gordon P Waldo, ed.-See NCJ-92338). [https://www.ncjrs.gov/App/Publications/abstract.aspx?ID=92340]

Lensvelt-Mulders, G. J., Hox, J. J., Van der Heijden, P. G., & Maas, C. J. (2005). Meta-analysis of randomized response research: Thirty-five years of validation. Sociological Methods & Research, 33(3), 319-348.

Wimbush, J. C., & Dalton, D. R. (1997). Base rate for employee theft: Convergence of multiple methods. Journal of Applied Psychology, 82(5), 756.

Reviewer #2: Dear authors,

I had the opportunity to review the manuscript “Indirect questioning methods for sensitive survey questions: modelling criminal behaviours among a prison population” for PLOS ONE. I have some comments that you might consider for this manuscript.

The introduction is clear and manages to cover the topics studied in this manuscript.

Please move the participants’ information from page 9 (table 1) to the methods section, so the reader can have a better reading of the methods and results sessions.

Please add more information related to the construction of the survey. For example, was it online? Was any software used for the design? Was it self-administered?

Within the inclusion criteria, it was taken into account the general cognitive state?

In relation to the participants who had no schooling (n = 125), how did they answer the test? Was there a way to ensure they understood the survey questions?

My main concern is related to Research Question 3. Tables 3-7 show the regression coefficients (p’s <0.05). However, they do not show the final model quality and its ability to predict / infer. My suggestion is using some cross-validation techniques, such as k fold validation or other techniques. Also, the authors can use statistical learning strategies (e.g. OneR, SimpleCart, and REPTree algorithms) for feature selection to determine the best variables for classification and compared models fits using the classification metrics: True Positive Rate (TPR), False Positive Rate (FPR), Precision and Accuracy, etc.; and then use cross-validation technical. The model presented in the manuscript could have an overfitting effect and not predict in a new sample / population.

Please clarify the alpha value used for the analyzes performed. Was Bonferroni correction used?

*Minor comments*

Please select the most representative graphics, I don't find very didactic to leave 15 similar graphics in a single manuscript.

Tables 3-7, please use scientific notation or use the expression p <0.001, for the p value = 0.000

6. PLOS authors have the option to publish the peer review history of their article (what does this mean?). If published, this will include your full peer review and any attached files.

Reviewer #1: No

Reviewer #2: No

---

## [Author Response · Author response to Decision Letter 0]

24 Nov 2020

Attached will you find the response to reviewer and editor comments

---

## [Decision Letter · Decision Letter 1]

4 Jan 2021

Indirect questioning methods for sensitive survey questions: modelling criminal behaviours among a prison population

PONE-D-20-23586R1

Dear Dr. Rueda,

We’re pleased to inform you that your manuscript has been judged scientifically suitable for publication and will be formally accepted for publication once it meets all outstanding technical requirements.

Kind regards,

José J. López-Goñi

Academic Editor

PLOS ONE

Additional Editor Comments (optional):

Thank you very much for your work and effort. As the reviewer has said, the main concerns have been addressed. Nice job!!

Reviewers' comments:

Reviewer's Responses to Questions

**Comments to the Author**

1. If the authors have adequately addressed your comments raised in a previous round of review and you feel that this manuscript is now acceptable for publication, you may indicate that here to bypass the “Comments to the Author” section, enter your conflict of interest statement in the “Confidential to Editor” section, and submit your "Accept" recommendation.

Reviewer #2: All comments have been addressed

2. Is the manuscript technically sound, and do the data support the conclusions?

Reviewer #2: Yes

3. Has the statistical analysis been performed appropriately and rigorously? 

Reviewer #2: Yes

4. Have the authors made all data underlying the findings in their manuscript fully available?

Reviewer #2: Yes

5. Is the manuscript presented in an intelligible fashion and written in standard English?

Reviewer #2: Yes

6. Review Comments to the Author

Reviewer #2: (No Response)

7. PLOS authors have the option to publish the peer review history of their article (what does this mean?). If published, this will include your full peer review and any attached files.

Reviewer #2: No

---

## [Editor Report · Acceptance letter]

8 Jan 2021

PONE-D-20-23586R1 

Indirect questioning methods for sensitive survey questions: modelling criminal behaviours among a prison population 

Dear Dr. Rueda:

I'm pleased to inform you that your manuscript has been deemed suitable for publication in PLOS ONE. Congratulations! Your manuscript is now with our production department. 

Kind regards, 

on behalf of

Dr. José J. López-Goñi 

Academic Editor

PLOS ONE